# Intra-Articular Injection of Adipose-Derived Stromal Vascular Fraction in Osteoarthritic Temporomandibular Joints: Study Design of a Randomized Controlled Clinical Trial

**DOI:** 10.3390/bioengineering11020171

**Published:** 2024-02-10

**Authors:** Jan Aart M. Schipper, Aartje Jorien Tuin, Joris A. van Dongen, Nico B. van Bakelen, Martin Conrad Harmsen, Fred K. L. Spijkervet

**Affiliations:** 1Department of Oral and Maxillofacial Surgery, University Medical Center Groningen, University of Groningen, 9713 Groningen, The Netherlands; a.j.tuin@umcg.nl (A.J.T.); n.b.van.bakelen@umcg.nl (N.B.v.B.); f.k.l.spijkervet@umcg.nl (F.K.L.S.); 2Department of Plastic Surgery, University Medical Center Utrecht, University of Utrecht, 3584 Utrecht, The Netherlands; jorisavandongen@gmail.com; 3Department of Pathology & Medical Biology, University Medical Center Groningen, University of Groningen, 9712 Groningen, The Netherlands; m.c.harmsen@umcg.nl

**Keywords:** tissue stromal vascular fraction, temporomandibular joint osteoarthritis, intra-articular injection, adipose tissue, adipose derived stromal cells

## Abstract

Introduction: Temporomandibular joint (TMJ) osteoarthritis is a degenerative disease of the TMJ. It is characterized by progressive degradation of the extracellular matrix components of articular cartilage, with secondary inflammatory components leading to pain in the temporomandibular region and reduced mouth opening. Current treatments do not halt disease progression, hence the need for new therapies to reduce inflammation and, consequently, improve symptoms. The aim of our randomized controlled clinical trial protocol is to investigate the efficacy of adjuvant intra-articular injections of autologous tissue-like stromal vascular fraction (tSVF), compared to arthrocentesis alone, in reducing pain and improving mouth opening in TMJ osteoarthritis patients. Materials and Methods: The primary endpoint analysis will consist of the visual analogue scale (VAS) for pain. The secondary endpoint analyses will include maximal interincisal mouth opening measurements; assessment of oral health and mandibular function based on the oral health impact profile (OHIP) questionnaire and mandibular functional impairment questionnaire (MFIQ); complications during the follow up; synovial cytokine analysis at baseline and after 26 weeks; and nucleated cells and tSVF (immuno)histochemistry analyses of the intervention group. Discussion: Our randomized clinical trial protocol will be applied to evaluate the efficacy of a new promising tSVF injection therapy for TMJ osteoarthritis. The safety of intra-articular injections of tSVF has been proven for knee osteoarthritis. However, since a tSVF injection is considered a heterologous application of cell therapy, the regulatory requirements are strict, which makes medical ethical approval challenging.

## 1. Introduction

Temporomandibular joint (TMJ) osteoarthritis is a degenerative disease of the TMJ. Women, especially menopausal women, are more likely to contract TMJ disorders than men (2:1) [1,2,3,4,5]. Several other non-inflammatory TMJ disorders, such as masticatory muscle pain or headaches, may also cause pain in the temporomandibular region and/or reduced mouth opening and thus demand appropriate diagnosis and therapy. The condition is characterized by progressive degradation of the extracellular matrix components of articular cartilage, with secondary inflammatory components leading to pain in the temporomandibular region and reduced mouth opening [6]. Multiple proteolytic enzymes such as proteases and cytokines (e.g., interleukin-1 (IL-1)) are involved in this cartilage degradation process [7].

The current management of TMJ osteoarthritis is based on non-invasive and invasive therapies. Current non-invasive therapies include occlusal splints, pain medication (such as NSAIDs, e.g., COX-2 inhibitors), as well as physical and psychological therapies [8,9,10]. If these non-invasive options do not reduce symptoms after 2–3 weeks, arthrocentesis, a minimally invasive therapy with or without corticosteroids/hyaluronate injections, should be considered. It involves the lavage of the upper or lower joint space using two communicating needles that are introduced into the upper compartment of the jaw. Invasive treatment options, including open joint surgeries, should be considered for severe cases with severe pathology, pain, and dysfunction [9]. A systematic review of randomized controlled trials of lavage therapy versus nonsurgical therapy in the treatment of arthralgia found that the former is slightly more effective at reducing pain [11]. A subsequent randomized controlled trial confirmed these results and even found that undertaking lavage therapy as the initial treatment reduces pain and functional impairment more rapidly than nonsurgical therapy [12,13]. The relative rapid improvement after arthrocentesis can be explained by the immediate removal of deleterious inflammatory cells and their secreted products such as pro-inflammatory cytokines, matrix-degrading enzymes, reactive oxygen species, and degeneration products, as well as molecules like prostaglandin E2 (PGE2). On the other hand, activated synoviocyte cells remain but might become quiescent due to the lack of chronic pro-inflammatory stimulation. Hence, arthrocentesis gives the joint a head start in the recuperation process. Arthrocentesis also increases mandibular mobility by removing intra-articular adhesions, eliminating the negative pressure in the joint, which reduces the mechanical obstruction [9,12]. While current treatments are symptom-based, the ultimate curative treatment would be to reverse the degeneration process, i.e., to regenerate the lost cartilage in the joint. However, such a curative treatment does not yet exist. The first step in these new therapies would be to stop the degenerative process. Inflammation has been correlated with the progression of TMJ osteoarthritis [14]. It is therefore necessary to develop an effective therapeutic agent that stops the degenerative process by suppressing inflammation and thus also reducing pain. Therapies halting disease progression could be especially effective in the early stages of TMJ osteoarthritis before most of the cartilage has been broken down.

While adipose tissue is already being used in open-joint TMJ reconstructions, a new autologous cell therapy for knee osteoarthritis, namely adipose-derived stromal vascular fraction (SVF) injections, has proved effective in improving pain and range of motion [15]. TMJ osteoarthritis and knee osteoarthritis are considered as low-inflammatory arthritic conditions, with increased pro-inflammatory cytokines in the synovial fluid such as IL-1beta, IL-6, TNF-alpha, and PGE_2_ (Figure 1) [9,16,17,18]. The presence of PGE_2_, one of the most important molecules that induces neuropathic pain [19,20], is more than 4-fold higher in TMJ than in knee osteoarthritis [18]. Synovial fluid analyses one year after an SVF injection showed reduced concentrations of pro-inflammatory molecules (matrix metalloproteinase-2 (MMP-2), IL1-beta, IL-6, and IL-8) and increased concentrations of anti-inflammatory molecules (insulin growth factor-1 (IGF-1) and IL-10) in knee osteoarthritis cases [21]. Anti-inflammatory therapy, such as SVF, is therefore needed for TMJ osteoarthritis to downregulate pro-inflammatory molecules, such as PGE_2_, to reduce pain symptoms. When adipose tissue is mechanically fractionated, this results in a tissue-like stromal vascular fraction (tSVF) [22]. tSVF has been extensively studied for its anti-inflammatory action [23]. In vitro TNFα-treated chondrocytes showed reduced IL-1β and COX2 gene expression after being co-cultured with tSVF-derived cells and conditioned medium [24]. A co-culture of SVF and TMJ-derived synoviocytes that had been exposed to osteoarthritis showed significantly more downregulated inflammatory molecules such as PGE_2_ and CXCL8/IL-8 compared to a synoviocyte monoculture [25].

Recently, the first two human case studies and a trial were published in which processed adipose tissue was injected into the joints of subjects with TMJ osteoarthritis and TMJ disorder [26,27,28]. These studies applied the processed adipose tissue using the nanofat procedure, the MyStem kit, and Lipogems technology, respectively [29,30,31]. They did not note any adverse events and reported reduced pain and improved mouth opening. However, the methodological quality of the two case studies was low because there were no controls. Our hypothesis is that using adjuvant tSVF with arthrocentesis therapy will inhibit the inflammatory response through the release of more anti-inflammatory factors than with arthrocentesis alone and that this will result in less pain and better mandibular function. A well-designed clinical trial is therefore warranted to investigate the effects of tSVF on TMJ osteoarthritis. Thus, the aim of this clinical trial is to investigate the efficacy of adjuvant autologous tSVF intra-articular injections compared to arthrocentesis alone to reduce pain and to increase mouth opening in TMJ osteoarthritis cases.

## 2. Methods and Design

### 2.1. Objectives

The overall aim of this prospective clinical trial is to investigate the efficacy of injecting tSVF intra-articularly to reduce pain and increase mouth function in TMJ osteoarthritis cases. The primary endpoint analysis will be the VAS. The secondary evaluated endpoints for the intervention group will be as follows: maximal interincisal mouth opening, oral health, and mandibular function based on the patient’s oral health impact profile (OHIP) questionnaire and the mandibular functional impairment questionnaire (MFIQ); complications during the follow-up; synovial cytokine analyses at baseline and after 26 weeks; and nucleated cells and SVF (immuno)histochemistry analyses.

### 2.2. Protocol and Registration

Our study is pending approval from the Dutch Central Committee for Clinical Research (CCMO).

### 2.3. Trial Design

This study is planned as a prospective double-blind randomized sham surgery controlled clinical trial. It will be an intervention study with a parallel design consisting of even treatment allocations (ratio of 1:1). All consecutive patients visiting the Department of Oral and Maxillofacial Surgery at the University Medical Center Groningen, The Netherlands, with osteoarthritis of the TMJ will be assessed for eligibility based on the inclusion and exclusion criteria mentioned in Table 1. It is estimated that acquiring the required number of patients suitable for inclusion will take around 1–2 years. The follow-up period will be 1-year post-intervention.

Patients will be included in the study after being evaluated by an oral and maxillofacial (OMF) surgeon who will then schedule them for baseline assessments by the investigator of pain, maximal interincisal opening (MIO), and collection of all the questionnaires (VAS, MFIQ, OHIP-49, SCL-90). The investigator will be blinded regarding treatment allocation during the entire study. After the baseline assessment, randomization will take place (minimization randomization; ALEA, Abcoude, The Netherlands). Both groups will be scheduled for an intervention at the Oral Maxillofacial Surgery outpatient clinic of the University Medical Center Groningen, The Netherlands. An envelope with the treatment allocation will only be handed to the OMF surgeon on the day of surgery.

The patients will not be informed about the treatment allocation. However, since the intervention group will receive an additional liposuction procedure, they could derive their treatment allocation from this additional procedure. Hence, a sham liposuction procedure will be performed on the other group’s patients by preparing a sterile surgical field on the abdomen, making a small stab incision, and then putting a medical plaster on the incision site. The control group patients will be informed about possible liposuction-related symptoms in the same manner as the intervention group patients receiving the actual liposuction. We will not perform the complete liposuction procedure in the control group for ethical reasons. Since the OMF surgeons perform the surgery, they will be informed about the treatment allocation.

Both groups’ post-operative follow-up examinations will be conducted at 3, 12, 26, and 52 weeks (Figure 2). Clinical care-as-usual examinations will be performed by the OMF surgeon to evaluate possible complications. Subsequently, measurements will be taken by the investigator in a separate consultation on the same days as the post-operative examinations.

### 2.4. Participants

In this study, we will ask patients to participate in our research after two consultations (Figure 3). The patients visiting the OMF surgery clinic with TMJ pain symptoms will first be given NSAIDs for 2 weeks. Based on the TC/TMD criteria, arthralgia can be diagnosed by both anamnestic history and exam [32]. To confirm arthralgia, a diagnostic anaesthetic (Ultracaïn forte, 1 mL, Aventis Pharma, Hoevelaken, The Netherlands) will be injected locally into the TMJ. If the pain disappears after this injection, it will be confirmation of TMJ intra-articular pain [33]. Hence, only those with a nociceptive temporomandibular pain disorder will be included in this study. Patients with chronic pain disorders, neuropathic pain, and parasympathetic pain will not respond to the local anaesthetic injection, and so will not be included in this study. At the subsequent 2-week consultation, the responsive patients will be informed about our prospective clinical study and asked if they want to participate. The patient will have two weeks to consider participating.

### 2.5. Surgical Procedure to Obtain and Inject Autologous tSVF

The intervention patient’s donor site (abdomen) will be infiltrated with tumescent solution (5 mL lidocaine 2% in 45 mL of lactated Ringer solution). In total, around 20 mL of lipoaspirate will be harvested from the lateral suprapubic abdominal region by applying manual suction pressure on a Sorensen cannula (Tulip Medical, San Diego, CA, USA) to generate 2 mL tSVF. One millilitre of the solution will be injected into the TMJ upper joint space while the other one millilitre will be subjected to laboratory analyses for quality control purposes.

The fractionation of adipose tissue (FAT) procedure will be executed according to the methods in our previous publication [34]. The procedure will be performed in one of the outpatient clinic treatment rooms of the department of Oral and Maxillofacial surgery at the University Medical Centre Groningen, The Netherlands. A biological safety cabinet will be used to process the tSVF harvested from the patient’s adipose tissue.

Under local anaesthesia, after locating the superior joint space, an 18-gauge needle will be introduced into all the patients’ mandibles at the level of the articular fossa. Usually, the patient senses a forward pressure in the mandible confirming the needle is in the desired location. This can be confirmed further by the backwash principle: fluid coming out of the joint space (i.e., for aspiration purposes from the joint space). Next, a maximum amount of 2 mL saline solution will be injected into the superior joint space, whereupon the patient will be asked to protrude and lateralize the mandible as much as possible and then to try and close the mouth. The syringe will be disconnected from the needle so as to eject the aspired saline solution from the syringe to send for laboratory analysis to determine cytokine concentrations. Then, more saline solution will be introduced into the superior joint space, raising the intra-articular pressure, and the patient will be asked to open their mouth slowly. Subsequently, the syringe will be disconnected from the needle, and the patient will have to close their mouth, which will cause the saline to be ejected through the needle. This step will be repeated two to three times, whereupon the patients’ mandible will be manipulated in forced maximal opening [35]. Finally, 1 mL of the intervention group patients’ own tSVF will be inserted through the same needle into their articular upper space. The control group patients will receive 1 mL of saline.

### 2.6. Outcome Measurements

Pain will be measured with the VAS. Secondary analysis will be performed by measuring mouth opening with a calliper by measuring maximal interincisal opening of the mandibular handicap with the Mandibular Functional Impairment Questionnaire (MFIQ) [36]. Oral health will be evaluated by the Oral Health Impact profile (OHIP-49) questionnaire [37]. The Symptom Check List (SCL-90) will be used to evaluate psychological well-being [38].

From each patient, 1 mL of tSVF will be fixed in 2% paraformaldehyde (PFA) for histological quality control of tSVF. Thin sections (4 µm) of paraffin-embedded tSVF will be deparaffinized and stained with Masson’s Trichrome, Safarin O, and Toluidine staining. Immunohistochemical staining will be performed for caspase 1, perilipin A, and vWF.

At baseline, synovial fluid will be collected during arthrocentesis, and after 26 weeks, an additional arthrocentesis will be performed to acquire synovial fluid for analysis in both the control and intervention group. Synovial fluid will be analysed with a Luminex assay containing pro- and anti-inflammatory factors, including TNF-alpha, IL-1beta, IL-6, IL-8, IFN-gamma, PGE_2,_ FGF, VEGF, IGF, HGF, and MMP-2.

### 2.7. Safety

Risks can be related to the surgical procedure or related to the injected product.

The injection of tissue into the intra-articular space can inflict the risk of infection. However, we will minimize procedure-related risks by producing tSVF in a biological safety cabinet. We will also conduct sterility testing on part of the tSVF that is prepared for injection. If the initial SVF was contaminated, we will then strictly follow these patients. If during the liposuction or isolation procedure of SVF the injected product becomes contaminated, this can lead to a septic arthritis of the TMJ. We have tested our FAT procedure in the operating room, and we have found no contamination due to our procedure (see IMPD). Septic arthritis can be managed by a regimen of non-steroidal anti-inflammatory agents, antibiotics, and arthrocentesis [39]. During the arthrocentesis, the infected tissue can be removed and will be sent to the laboratory for microbial analysis. After this tissue has been removed, the joint can be flushed with saline to alleviate any remaining pressure. When the microbial analysis is finished, a more specific antibiotic therapy can be started. The risk of facial nerve injury in arthrocentesis of the TMJ joint is extremely small [40]. The procedure of injecting adipose tissue is performed through the same needle used during the arthrocentesis. Therefore, the SVF-procedure will not pose any additional risk of nerve injury compared to standard care.

Risks due to the injected product are not expected but can occur because the mechanism of action of SVF is only partly known. Because SVF is such a heterogeneous mixture of cells and the extracellular matrix, its effect is most probably achieved through a variety of mechanisms. The progenitor cells in the tissue can differentiate into cartilage. But the progenitor cells can also have their effect through a paracrine mechanism by excreting anti-inflammatory cytokines, growth factors, or exosomes. It is known that SVF reduces the amount of IL-6 and TNF-alpha [41]. In addition to this, adipose tissue can function as a mechanical protective layer and lubricant, in the same way adipose tissue is used in temporomandibular surgery at the moment. Since the mechanism is not fully known, it theoretically poses the risk of carcinogenesis. The addition of growth factors or immunomodulation is of course reason for concern when there is a chance of tissue differentiating to neoplasms. It is also possible that the immunosuppressive effects at the site of implantation stimulate pre-existing tumours. However, only in vitro studies with expanded ASCs with pre-existing tumour cells have shown that tumour growth is stimulated [42]. We will use non-expanded adipose tissue. In addition to this, tumours of the TMJ are very rare [43]. Therefore, the potential risk of the development of neoplasms is considered negligible.

### 2.8. Sample Size Calculation

To the best of our knowledge, no (pilot) studies have been performed comparing the efficacy of tSVF with only arthrocentesis to treat osteoarthritis of the TMJ. Therefore, we performed a sample size calculation based on the assumption that the difference in mean change between the interventions would be 0.8 for our primary outcome variable (pain during movement).

Regarding the sample size calculation, the following values were used for a longitudinal random effects analysis: α = 0.05, power = 0.80, and an allocation ratio of 1:1 (r = 1). A formula was used from the literature [44], whereby Z(1 − α/2) = 1.96, Z(1 − β) = 0.84, σ = 0.94 (the pooled SD is based on the baseline measurements De Riu [45]), T = 4 follow-up moments, ρ = 0.7 (our estimation of the correlation between observations over time), v = 0.8 (difference in means). This results in a sample size of 16.8 ≈ 17 per group. To compensate for loss to follow-up, we plan to include 20 participants in each group.

### 2.9. Statistical Analysis

Since we are interested in the development of the difference in outcome variables between the groups over time (1 year) and because we assume that missing values may occur due to participants missing appointments, we plan to perform a longitudinal random effects analysis for the primary endpoint. In the secondary endpoint analysis, we will analyse the main effects of group and time as well as the interaction effect of group and time. Differences in these scores between the treatment and control groups will be analysed by the longitudinal random effects model. Cytokine levels will be related to outcome measures of the study by calculating the Pearson correlation coefficient for each outcome measure. Descriptive statistics will be provided for all outcome measurements.

## 3. Discussion

The aim of this double-blind randomized sham surgery controlled clinical trial is to investigate the efficacy of an intra-articular injection of tSVF, as an adjuvant to arthrocentesis, in reducing pain and increasing mouth opening in TMJ osteoarthritis patients.

A tSVF injection in the TMJ is categorized as cellular therapy, which has consequences on its medical ethical approval and its regulatory burden. tSVF obtained through mechanical fractionation can be considered as being minimally manipulated, contrary to enzymatically digested cellular stromal vascular fractions (cSVFs). The European Medicines Agency considers a product not substantially manipulated when it ‘does not contain cells that have been subject to substantial manipulation so that biological characteristics, physiological functions, or structural properties relevant for the intended clinical use or for the intended regeneration, repair or replacement have been altered’ [46]. However, since we will inject adipose-derived SVFs into the donor patient’s TMJ space, it will not contain ‘cells that are intended to be used for the same essential function(s) in the recipient and the donor’ [46]. This heterologous application categorizes tSVF as a cellular therapy, i.e., an advanced therapy medicinal product (ATMP). Although the safety of intra-articular SVF injections has been proven and well-documented [47,48,49], national medical ethical committee approval can only be obtained by strictly adhering to the rules for cellular therapy products. Quality control is essential since tSVF composition is variable between donors, and this can influence the treatment’s effect. However, tSVF is a heterogenous product with an intact extracellular matrix, which makes it difficult to characterize with conventional single-cell suspension characterization assays since the cells are bound to the extracellular matrix. The tSVF will be injected during the same surgical procedure immediately after fractionation, which makes it impossible to undertake a quality control before the injection. Medical ethical approval demands testing of whether tSVF components are present in pre-defined concentrations. This criterion implies, however, that certain separate components of tSVF are responsible for its clinical efficacy. Yet it is also possible that the synergy of the components of tSVF is responsible for its clinical efficacy. Nevertheless, at this point in time, we do not know whether specific tSVF components are responsible for its mechanism of action, i.e., the suppression of inflammation, or which components those might be. Hence, it is impossible to determine the acceptable concentration range of the tSVF components to pass the quality control. In addition to this, since the tSVF will be injected intra-operatively immediately after fractionation, controlling its quality will only be possible after being injected into the TMJ, while most cellular ATMPs need to be quality-assessed before administration.

The SVF is obtained through adipose tissue processing. Adipose tissue harvested through a liposuction procedure consists of two fractions: adipocytes and SVF. tSVF is produced through mechanical fractionation by forcing the adipose tissue through a device with a small hole and centrifuging it twice to disrupt the adipocytes and remove the oil [22]. Our recent scoping review showed that the first centrifugation step is especially essential to remove oil from the adipose tissue by adequately disrupting the adipocytes to release their triglycerides [22]. In a trial, we used the FAT procedure to yield tSVF consisting of a heterogeneous mixture of endothelial cells, pericytes, preadipocytes, fibroblasts, macrophages, and adipose-derived stromal cells (ASC), while preserving the intercellular connections, including the extracellular matrix [50]. The extracellular matrix functions as a slow-release platform for factors secreted by these cells and probably results in longer cell survival [51].

The hypothesized mechanism of action of tSVF in a chronically low-grade-inflamed osteoarthritic joint is that the cells secrete anti-inflammatory factors in a paracrine fashion, which downregulates inflammation. Since the cells will probably not be engrafted in the joint, cartilage regeneration is not expected to occur. Testing this hypothesis is necessary to understand its mechanism of action better. We will therefore test the synovial fluid at both baseline and after 26 weeks for pro-inflammatory and anti-inflammatory factors in both the intervention and the control group. We expect that tSVF will reduce potentially harmful factors such as IL-1B, TNF-alpha, chemoattractants, and MMPs, as well as PGE2, one of the most important factors that maintains the chronic inflammation in the joint. In addition to this, we pose that tSVF, through the secretion of growth factors such as FGF, VEGF, IGF, and HGF, sets the local micro-environment into ‘regenerative mode’ by putting activated synoviocytes into rest mode. There is no synovial fluid analysis data available from TMJ osteoarthritis patients after a tSVF injection. However, in a study of patients with knee osteoarthritis, a reduction in pro-inflammatory molecules (MMP-2, IL-1B, IL-6, and IL-8) and an increase in anti-inflammatory molecules (IGF-1 and IL-10) were found at the 1-year follow-up after an SVF injection [21]. However, performing an additional arthrocentesis for synovial fluid analysis in the recovering joint after 26 weeks to test the mechanism of action for scientific reasons could be considered unethical since there is no benefit for the patient. Nonetheless, because these adipose-derived regenerative treatments are still in their early stages, testing their mechanisms of action is often required to obtain medical ethical approval.

There are some limitations to this trial: we will not investigate which tSVF dosage should be injected, the number of injections that would be sufficient, whether differences in patient characteristics influence the efficacy of autologous tSVF, nor whether the stage of the disease influences the treatment potential. We chose to use sham surgery in the form of making a small stab incision in the control group in this study design. However, since no bruising will occur because no liposuction will be performed, the patients will be able to deduce their treatment allocation. We decided not to perform a complete liposuction procedure in the control group because this would impose unnecessary treatment-related side effects.

The first two human trials of injecting adipose-derived components showed a reduction in pain and increased mouth opening [26,27]. However, the methodological quality of these non-controlled trials was low. A recent randomized controlled trial compared a control arthrocentesis group with an experimental group injected with micro-fragmented adipose tissue obtained by the Lipogems procedure [28]. This method leads to adipose cluster reduction by using metal balls in a device which does not disrupt all the adipocytes and is therefore not similar to our FAT procedure, which produces tSVF [52]. Both pain (VAS) and mouth function (interincisal opening) were measured preoperatively and during the follow-ups 10 days, 1 month, and 6 months after the injection. The trial reported that the experimental group had a notable reduction in pain at both the 10-day and 1-month follow-ups but the mouth opening only began to improve 6 months after the injection. This result suggests that adipose tissue injection leads to immediate pain reduction, but mouth opening only improves after 6 months. Pain reduction and functional improvement have also been seen with osteoarthritic knees following this procedure [15,53,54,55]. To our knowledge, there are no animal trials describing the injection of tSVF in TMJ osteoarthritis models. However, animal trials have shown increased cartilage thickness after adipose-derived stromal cell injections compared to control groups, a finding that has not been confirmed in human trials [56]. The safety of intra-articular injections in knee osteoarthritis has been well proven. A recent systematic review noted few minor complications and one major complication after 4008 knee injections; there was one case of knee joint infection which was treated successfully by means of a synovectomy and antibiotics [49]. Following 264 intra-articular injections, only 7 minor complications and no serious adverse events were reported [47]. After 480 knee injections, one patient had symptoms of transient pain and swelling, but the symptoms resolved spontaneously and subsequent injections were uneventful [57].

In conclusion, our double-blind randomized sham surgery controlled clinical trial will investigate whether a tSVF injection, as an adjuvant to arthrocentesis, will reduce the symptoms in TMJ osteoarthritis patients and increase their mouth opening. This may potentially lead to new treatment options for TMJ osteoarthritis.

## Figures and Tables

**Figure 1 bioengineering-11-00171-f001:**
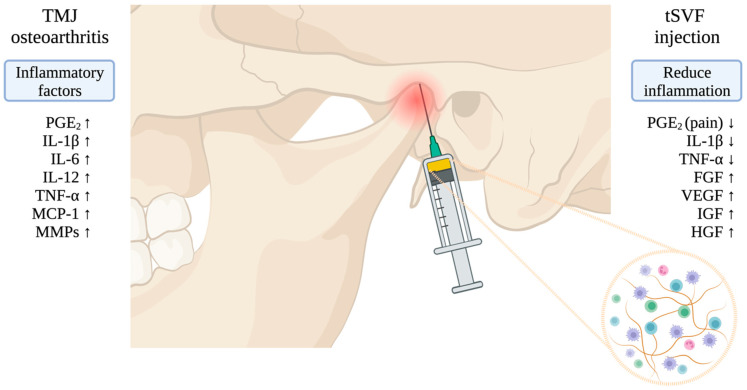
The pathogenesis and hypothesized mechanism of action of tSVF injection on TMJ osteoarthritis. (arrow up means higher concentration; arrow down means lower concentration).

**Figure 2 bioengineering-11-00171-f002:**
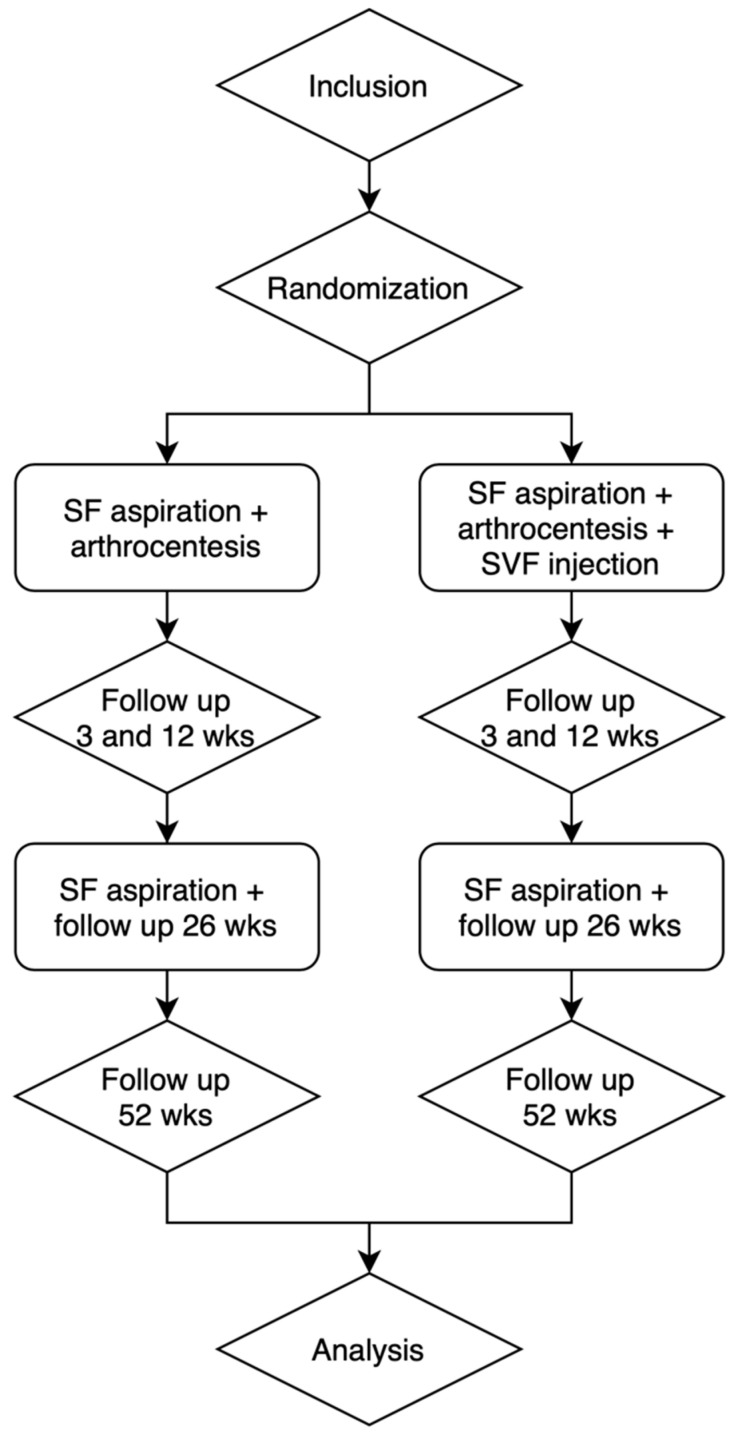
Follow-up.

**Figure 3 bioengineering-11-00171-f003:**
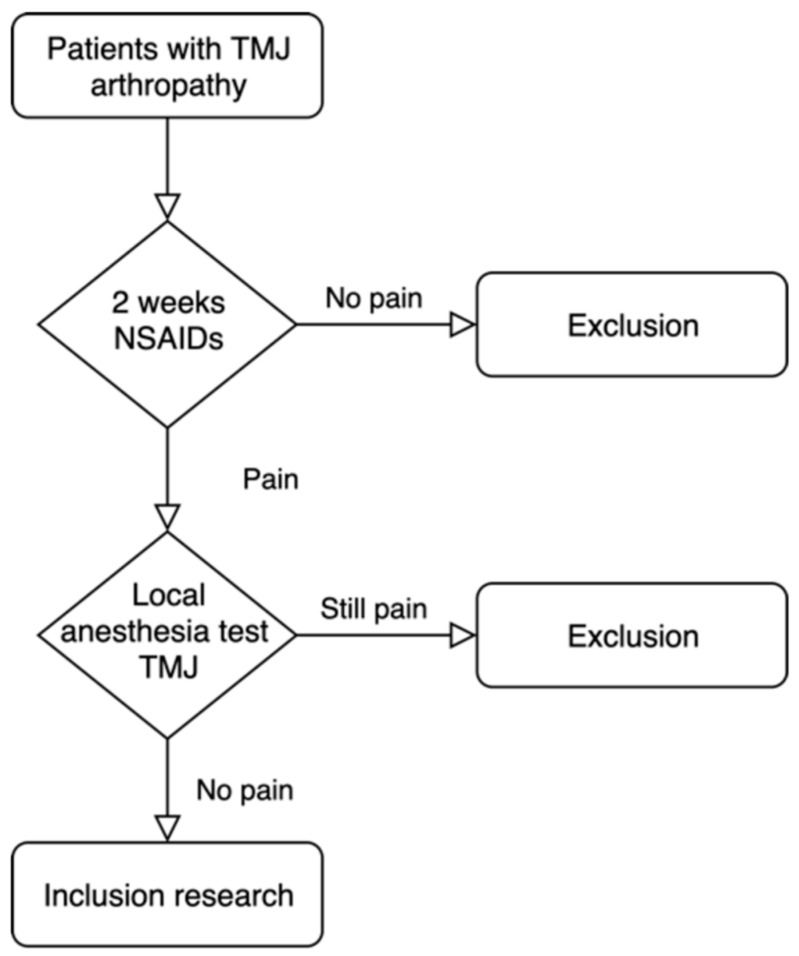
Inclusion test.

**Table 1 bioengineering-11-00171-t001:** Inclusion and exclusion criteria.

Inclusion	Exclusion
Age between 18 and 70 years Chronic nociceptive pain in the TMJ region, aggravated by protrusion, maximal mouth opening, lateral excursions, and/or chewing for at least 2 months Wilkes stages III or IV (internal derangement) Limited maximal interincisal opening (<35 mm and >15 mm) Pain still present after two weeks of an NSAID (i.e., ibuprofen 600 mg three times daily or diclophenac 50 mg 3 td or naproxen 500 mg 2 td) Pain disappears after diagnostic intra-articular injection of local anesthetic	Edentulous (no dentition) Concurrent use of anti-inflammatory medication, steroids, muscle relaxants, or antidepressants Previous TMJ traumas and fractures Previous TMJ surgeries, previous intra-articular injections within <1 year Previous osteotomies of the mandible Bilateral severe TMJ derangements Bony or fibrotic ankylosis of the TMJ Known history of diabetes mellitus type 1 or 2 Known history of HIV Known history of Hepatitis B or C Serious systemic diseases, rheumatic disease, or infectious/inflammatory diseases affecting the skin of the TMJ area Pregnancy Coagulation disorders BMI < 15

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
