# Peer review of "Intra-Articular Injection of Adipose-Derived Stromal Vascular Fraction in Osteoarthritic Temporomandibular Joints: Study Design of a Randomized Controlled Clinical Trial"

_bioengineering, 2024, doi:10.3390/bioengineering11020171_

Round 1
Reviewer 1 Report
Comments and Suggestions for Authors
The present manuscript titled: ‘Intra-articular injection of adipose-derived stromal vascular fraction in osteoarthritic temporomandibular joints: study design of a randomized controlled clinical trial.’ is well written and a protocol of a very interesting randomized controlled trial. However, I do have a few remarks to be considered before publication.
As regulatory requirements are very strict and as the authors already mentioned in their last sentence of the abstract, I would point out that this kind of treatment will be performed as a single procedure of intra-articular injection of autologous adipose-derived stromal vascular fraction. Despite challenging requirements for ATMPs, doctors must ask themselves if they are performing a surgical treatment or are they producing a medicinal product (drug), which is industrially produced and that is then to be brought onto the market… Thus, it is very important to make clear that the planned procedure will be performed as an autologous procedure and hence, that (as mentioned in the manuscript on page 9 first paragraph) tSVF components of different patients (they also will vary in one patient at different time points!) will never ever be ‘produced’ in a pre-defined concentration. A patient is not device….
-I would suggest that you mention the ‘autologous’ type of procedure throughout the protocol more often. Maybe already in the title, but definitely already in the abstract.
Another important fact to be considered in fairly new methods is safety. Therefore, it is advisable to capture and report adverse events (AEs)/complications in depth.
-Please include further details regarding AEs in the method section.
There are two primary endpoints (pain as short-term and mouth opening as later outcome). From the statistical point of view, if using multiple primary outcomes, one must consider correction. The likelihood of making false conclusions about an effect when using more than one endpoints becomes a concern if there is not appropriate adjustment for multiplicity.
-Hence, either you consider corrections or use another endpoint. Maybe one of your scores includes pain and function...
-Please include your statistical analysis plan!
Please find further detailed comments and suggestions in the pdf file!

Comments on the Quality of English LanguageAuthor Response
Please see attachment.

Reviewer 2 Report
Comments and Suggestions for Authors
The authors of the study "Intra-articular injection of adipose-derived stromal vascular fraction in osteoarthritic temporomandibular joints: study design of a randomized controlled clinical trial" present a very interesting protocol to evaluate the potential of tSVF in cases of osteoarthritis. The study is well-designed. I only have a few recommendations and concerns:
- The authors should make one last revision in terms of English and the uniformity of the numbers in full or numerical numbers or abbreviations, namely right in the abstract in this sentence: "Temporomandibular joint (TMJ) osteoarthritis is a degenerative disease of the temporomandibular joint".
- What is the rationale for taking the sample for protein analysis at 26 weeks?
- The authors should incorporate a psychological analysis of the patient's GAD and PHQ, bearing in mind that the primary outcome is VAS pain. Anxiety and depression could be considered a confounding agent of the results and help in the analysis.
- What is the explanation for the inclusion criteria: "Pain disappears after diagnostic intra-articular injection"?
- The authors should also include which diagnostic method they are going to use to diagnose osteoarthritis and authors should follow the DC/TMD recommendations.
Comments on the Quality of English LanguageA final review is needed.
Reviewer 3 Report
Comments and Suggestions for Authors
In this manuscript, authors designed a prospective double-blind, randomized, controlled clinical trial to investigate the efficacy of an intra-articular injection of tissue-like stromal vascular fraction (tSVF), as an adjuvant to arthrocentesis, in reducing pain and increasing mouth opening in TMJ osteoarthritis patients. The clinical trial is well designed and protocol to be followed is well explained in the manuscript. The trial design methodology including objectives, no. of patients, inclusion / exclusion criteria, protocols and registration, outcomes of the trial are written in details in the manuscript.
However, manuscript represents a final protocol for a prospective clinical trial to be performed in the future. Patients have not been recruited yet in this trial and hence lack of any data / results in the manuscript. In the absence of the results, conclusions are based on hypothesis only. Although interesting in its present format, this manuscript should be published with results and final conclusion based on the data.
Round 2
Reviewer 2 Report
Comments and Suggestions for Authors
The authors answered all the questions.
Reviewer 3 Report
Comments and Suggestions for Authors
None..